# Partition-wise Linear Models

**Hidekazu Oiwa**[*]
Graduate School of Information Science and Technology
The University of Tokyo
hidekazu.oiwa@gmail.com

**Ryohei Fujimaki**
NEC Laboratories America
rfujimaki@nec-labs.com

## Abstract

Region-specific linear models are widely used in practical applications because of their non-linear but highly interpretable model representations. One of the key challenges in their use is non-convexity in simultaneous optimization of regions and region-specific models. This paper proposes novel convex region-specific linear models, which we refer to as partition-wise linear models. Our key ideas are 1) assigning linear models not to regions but to partitions (region-specifiers) and representing region-specific linear models by linear combinations of partition-specific models, and 2) optimizing regions via partition selection from a large number of given partition candidates by means of convex structured regularizations. In addition to providing initialization-free globally-optimal solutions, our convex formulation makes it possible to derive a generalization bound and to use such advanced optimization techniques as proximal methods and decomposition of the proximal maps for sparsity-inducing regularizations. Experimental results demonstrate that our partition-wise linear models perform better than or are at least competitive with state-of-the-art region-specific or locally linear models.

## 1 Introduction

Among pre-processing methods, data partitioning is one of the most fundamental. In it, an input space is divided into several sub-spaces (regions) and assigned a simple model for each region. In addition to better predictive performance resulting from the non-linear nature that arises from multiple partitions, the regional structure provides a better understanding of data (i.e., interpretability). Region-specific linear models learn both partitioning structures and predictors in each region.

Such models vary—from traditional decision/regression trees [1] to more advanced models [2, 3, 4]—depending on their region-specifiers (how they characterize regions), region-specific prediction models, and the objective functions to be optimized. One important challenge that remains in learning these models is the non-convexity that arises from the inter-dependency of optimizing regions and prediction models in individual regions. Most previous work suffers from disadvantages arising from non-convexity, including initialization-dependency (bad local minima) and lack of generalization error analysis.

We propose convex region-specific linear models, which are referred to as partition-wise linear models. Our models have two distinguishing characteristics that help avoid the non-convexity problem.

**Partition-wise Modeling**   We propose partition-wise linear models as a novel class of region-specific linear models. Our models divide an input space by means of a small set of partitions[1]. Each partition possesses one weight vector, and this weight vector is only applied to one side of the divided space. It is trained to represent the local relationship between input vectors and output

---

[*]The work reported here was conducted when the first author was a visiting researcher at NEC Laboratories America.

[1]In our paper, a region is a sub-space in an input space. Multiple regions do not intersect each other, and, in their entirety, they cover the whole input space. A partition is an indicator function that divides an input space into two parts.

values. Region-specific predictors are constructed by linear combinations of these weight vectors. Our partition-wise parameterization enables us to construct convex objective functions.

**Convex Optimization via Sparse Partition Selection**   We optimize regions by selecting effective partitions from a large number of given candidates, using convex sparsity-inducing structured regularizations. In other words, we trade continuous region optimization for convexity. We allow partitions to locate only given discrete candidate positions, and are able to derive convex optimization problems. We have developed an efficient algorithm to solve structured-sparse optimization problems, and in it we adopt a proximal method [5, 6] and the decomposition of proximal maps [7].

As a reliable partition-wise linear model, we have developed a global and local residual model that combines one global linear model and a set of partition-wise linear ones. Further, our theoretical analysis gives a generalization bound for this model to evaluate the risk of over-fitting. Our generalization bound analysis indicates that we can increase the number of partition candidates by less than an exponential order with respect to the sample size, which is large enough to achieve good predictive performance in practice. Experimental results have demonstrated that our models perform better than or are at least competitive with state-of-the-art region-specific or locally linear models.

## 1.1   Related Work

Region-specific linear models and locally linear models are the most closely related models to our own. The former category, to which our models belong, assumes one predictor in a specific region and has an advantage in clear model interpretability, while the latter assigns one predictor to every single datum and has an advantage in higher model flexibility. Interpretable models are able to indicate clearly where and how the relationships between inputs and outputs change.

Well-known precursors to region-specific linear models are decision/regression trees [1], which use rule-based region-specifiers and constant-valued predictors. Another traditional framework is a hierarchical mixture of experts [8], which is a probabilistic tree-based region-specific model framework. Recently, Local Supervised Learning through Space Partitioning (LSL-SP) has been proposed [3]. LSL-SP utilizes a linear-chain of linear region-specifiers as well as region-specific linear predictors. The highly important advantage of LSL-SP is the upper bound of generalization error analysis via the VC dimension. Additionally, a Cost-Sensitive Tree of Classifiers (CSTC) algorithm has also been developed [4]. It utilizes a tree-based linear localizer and linear predictors. This algorithm's uniqueness among other region-specific linear models is in its taking "feature utilization cost" into account for test time speed-up. Although the developers' formulation with sparsity-inducing structured regularization is, in a way, related to ours, their model representations and, more importantly, their motivation (test time speed-up) is different from ours.

Fast Local Kernel Support Vector Machines (FaLK-SVMs) represent state-of-the-art locally linear models. FaLK-SVMs produce test-point-specific weight vectors by learning local predictive models from the neighborhoods of individual test points [9]. It aims to reduce prediction time cost by preprocessing for nearest-neighbor calculations and local model sharing, at the cost of initialization-independency. Another advanced locally linear model is that of Locally Linear Support Vector Machines (LLSVMs) [10]. LLSVMs assign linear SVMs to multiple anchor points produced by manifold learning [11, 12] and construct test-point-specific linear predictors according to the weights of anchor points with respect to individual test points. When the manifold learning procedure is initialization-independent, LLSVMs become initial-value-independent because of the convexity of the optimization problem. Similarly, clustered SVMs (CSVMs) [13] assume given data clusters and learn multiple SVMs for individual clusters simultaneously. Although CSVMs are convex and generalization bound analysis has been provided, they cannot optimize regions (clusters).

Joes et al. have proposed Local Deep Kernel Learning (LDKL) [2], which adopts an intermediate approach with respect to region-specific and locally linear models. LDKL is a tree-based local kernel classifier in which the kernel defines regions and can be seen as performing region-specification. One main difference from common region-specific linear models is that LDKL changes kernel combination weights for individual test points, and the predictors are locally determined in every single region. Its aim is to speed up kernel SVMs' prediction while maintaining the non-linear ability.

Table 1 summarizes the above described state-of-the-art models in contrast with ours from a number of significant perspectives. Our proposed model uniquely exhibits three properties: joint optimization of regions and region-specific predictors, initialization-independent optimization, and meaningful generalization bound.

Table 1: Comparison of region-specific and locally linear models.

| | Ours | LSL-SP | CSTC | LDKL | FaLK-SVM | LLSVM |
|---|---|---|---|---|---|---|
| Region Optimization | √ | √ | √ | √ | | |
| Initialization-independent | √ | | | | | √ |
| Generalization Bound | √ | √ | | | √ | |
| Region Specifiers | (Sec. 2.2) | Linear | Linear | Linear | Non-Regional | Non-Regional |

## 1.2 Notations

Scalars and vectors are denoted by lower-case $x$. Matrices are denoted by upper-case $X$. An $n$-th training sample and label are denoted by $x_n \in \mathbb{R}^D$ and $y_n$, respectively.

## 2 Partition-wise Linear Models

This section explains partition-wise linear models under the assumption that effective partitioning is already fixed. We discuss how to optimize partitions and region-specific linear models in Section 3.

### 2.1 Framework

Figure 1 illustrates the concept of partition-wise linear models. Suppose we have $P$ partitions (red dashed lines) which essentially specify $2^P$ regions. Partition-wise linear models are defined as follows. First, we assign a linear weight vector $a_p$ to the $p$-th partition. This partition has an *activeness function*, $f_p$, which indicates whether the attached weight vector $a_p$ is applied to individual data points or not. For example, in Figure 1, we set the weight vector $a_1$ to be applied to the right-hand side of partition $p_1$. In this case, the corresponding activeness function is defined as $f_1(x) = 1$ when $x$ is in the right-hand side of $p_1$. Second, region-specific predictors (squared regions surrounded by partitions in Figure 1) are defined by a linear combination of active partition-wise weight vectors that are also linear models.

Let us formally define the partition-wise linear models. We have a set of given activeness functions, $f_1, \ldots, f_P$, which is denoted in a vector form as $f(\cdot) = (f_1(\cdot), \ldots, f_P(\cdot))^T$. The $p$-th element $f_p(x) \in \{0, 1\}$ indicates whether the attached weight vector $a_p$ is applied to $x$ or not. The activeness function $f(\cdot)$ can represent at most $2^P$ regions, and $f(x)$ specifies to which region $x$ belongs. A linear model of an individual region is then represented as $\sum_{p=1}^{P} f_p(\cdot)a_p$. It is worth noting that partition-wise linear models use $P$ linear weight vectors to represent $2^P$ regions and restrict the number of parameters.

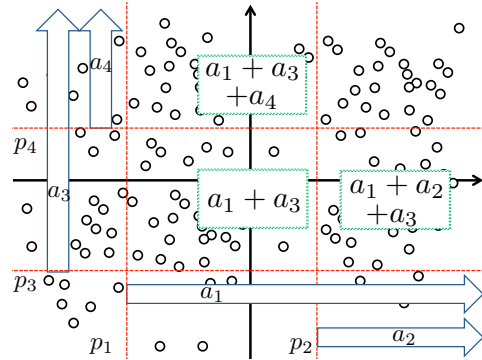

Figure 1: Concept of Partition-wise Linear Models

The overall predictor $g(\cdot)$ can be denoted as follows:

$$g(x) = \sum_p f_p(x) \sum_d a_{dp} x_d. \tag{1}$$

Let us define $A$ as $A = (a_1, \ldots, a_P)$. The partition-wise linear model (1) simply acts as a linear model w.r.t. $A$ while it captures the non-linear nature of data (individual regions use different linear models). Such non-linearity originates from the activeness functions $f_p$s, which are fundamentally important components in our models.

By introducing a convex loss function $\ell(\cdot, \cdot)$ (e.g., squared loss for regression, squared hinge or logistic loss for classification), we can represent an objective function of the partition-wise linear models as a convex loss minimization problem as follows:

$$\min_A \sum_n \ell(y_n, g(x_n)) = \min_A \sum_n \ell(y_n, \sum_p f_p(x_n) \sum_d a_{dp} x_{nd}). \tag{2}$$

Here we give a convex formulation of region-specific linear models under the assumption that a set of partitions is given. In Section 3, we propose a convex optimization algorithm for partitions and regions as a partition selection problem, using sparsity-inducing structured regularization.

## 2.2 Partition Activeness Functions

A partition activeness function $f_p$ divides the input space into two regions, and a set of activeness functions defines the entire region-structure. Although any function is applicable in principle to being used as a partition activeness function, we prefer as simple a region representation as possible because of our practical motivation of utilizing region-specific linear models (i.e., interpretability is a priority). This paper restricts them to being parallel to the coordinates, e.g., $f_p(x) = 1$ ($x_i > 2.5$) and $f_p(x) = 0$ (otherwise) with respect to the $i$-th coordinate. Although this "rule-representation" is simpler than others [2, 3] which use dense linear hyperplanes as region-specifiers, our empirical evaluation (Section 5) indicates that our models perform competitively with or even better than those others by appropriately optimizing the simple region-specifiers (partition activeness functions).

## 2.3 Global and Local Residual Model

As a special instance of partition-wise linear models, we here propose a model which we refer to as a global and local residual model. It employs a global linear weight vector $a_0$ in addition to partition-wise linear weights. The predictor model (1) can be rewritten as:

$$g(x) = a_0^T x + \sum_p f_p(x) \sum_d a_{dp} x_d \ . \tag{3}$$

The global weight vector is active for all data. The integration of the global weight vector enables the model to determine how features affect outputs not only locally but also globally. Let us consider a new partition activeness function $f_0(x)$ that always returns to 1 regardless of $x$. Then, by setting $f(\cdot) = (f_0(\cdot), f_1(\cdot), \ldots, f_p(\cdot), \ldots, f_P(\cdot))^T$ and $A = (a_0, a_1, \ldots, a_P)$, the global and local residual model can be represented using the same notation as is used in Section 2.1. Although $a_0$ and $a_p$ have no fundamental difference here, they are different in terms of how we regularize them (Section 3.1).

# 3 Convex Optimization of Regions and Predictors

In Section 2, we presented a convex formulation of partition-wise linear models in (2) under the assumption that a set of partition activeness functions was given. This section relaxes this assumption and proposes a convex partition optimization algorithm.

## 3.1 Region Optimization as Sparse Partition Selection

Let us assume that we have been given $P+1$ partition activeness functions, $f_0, f_1, \ldots, f_P$, and their attached linear weight vectors, $a_0, a_1, \ldots, a_P$, where $f_0$ and $a_0$ are the global activeness function and weight vector, respectively. We formulate the region optimization problem here as partition selection by setting setting most of $a_p$s to zero since $a_p = 0$ corresponds to the situation in which the $p$-th partition does not exist.

Formally, we formulate our optimization problem with respect to regions and weight vectors by introducing two types of sparsity-inducing constrains to (2) as follows:

$$\min_A \sum_n \ell(y_n, g(x_n)) \ \text{s.t.} \sum_{p \in \{1,\ldots,P\}} 1_{\{a_p \neq 0\}} \leq \mu_P, \ \|a_p\|_0 \leq \mu_0 \ \forall p. \tag{4}$$

The former constraint restricts the number of effective partitions to at most $\mu_P$. Note that we do not enforce this sparse partition constraint to the global model $a_0$ so as to be able to determine local trends as residuals from a global trend. The latter constraint restricts the number of effective features of $a_p$ to at most $\mu_0$. We add this constraint because 1) it is natural to assume only a small number of features are locally effective in practical applications and 2) a sparser model is typically preferred for our purposes because of its better interpretability.

## 3.2 Convex Optimization via Decomposition of Proximal map

### 3.2.1 The Tightest Convex Envelope

The constraints in (5) are non-convex, and it is very hard to find the global optimum due to the indicator functions and $L_0$ penalties. This makes optimization over a non-convex region a very complicated task, and we therefore apply a convex relaxation. One standard approach to convex relaxation would be a combination of group $L_1$ (the first constraint) and $L_1$ (the second constraint) penalties. Here, however, we consider the tightest convex relaxation of (4) as follows:

$$\min_A \sum_n \ell(y_n, g(x_n)) \ \text{s.t.} \ \sum_{p=1}^{P} \|a_p\|_\infty \leq \mu_P, \ \sum_{d=1}^{D} \|a_{dp}\|_\infty \leq \mu_0 \ \forall p. \tag{5}$$

The tightness of (5) is shown in the full version [14]. Through such a convex envelope of constraints, the feasible region becomes convex. Therefore, we can reformulate (5) to the following equivalent problem:

$$\min_A \sum_n \ell(y_n, g(x_n)) + \Omega(A) \text{ where } \Omega(A) = \lambda_P \sum_{p=1}^{P} \|a_p\|_\infty + \lambda_0 \sum_{p=0}^{P} \sum_{d=1}^{D} \|a_{dp}\|_\infty, \quad (6)$$

where $\lambda_P$ and $\lambda_0$ are regularization weights corresponding to $\mu_P$ and $\mu_0$, respectively. We derive an efficient optimization algorithm using a proximal method and the decomposition of proximal maps.

### 3.2.2 Proximal method and FISTA

The proximal method is a standard efficient tool for solving convex optimization problems with non-differential regularizers. It iteratively applies gradient steps and proximal steps to update parameters. This achieves $O(1/t)$ convergence [5] under Lipschitz-continuity of the loss gradient, or even $O(1/t^2)$ convergence if an acceleration technique, such as a fast iterative shrinkage thresholding algorithm (FISTA) [6, 15], is incorporated.

Let us define $A^{(t)}$ as the weight matrix at the $t$-th iteration. In the gradient step, the weight vectors are updated to decrease empirical loss through the first-order approximation of loss functions as:

$$A^{(t+\frac{1}{2})} = A^{(t)} - \eta^{(t)} \sum_n \partial_{A^{(t)}} \ell(y_n, g(x_n)), \quad (7)$$

where $\eta^{(t)}$ is a step size and $\partial_{A^{(t)}} \ell(\cdot, \cdot)$ is the gradient of loss functions evaluated at $A^{(t)}$. In the proximal step, we apply regularization to the current solution $A^{(t+\frac{1}{2})}$ as follows:

$$A^{(t+1)} = M_0(A^{(t+\frac{1}{2})}) \text{ where } M_0(B) = \underset{A}{\operatorname{argmin}} \left( \frac{1}{2} \|A - B\|_F^2 + \eta^{(t)} \Omega(A) \right), \quad (8)$$

where $\| \cdot \|_F$ is the Frobenius norm. Furthermore, we employed FISTA [6] to achieve the faster convergence rate for weakly convex problem and adopted a backtracking rule [6] to avoid the difficulty of calculating appropriate step widths beforehand. Through empirical evaluations as well as theoretical backgrounds, we have confirmed that it significantly improves convergence in learning partition-wise linear models. The detail is written in the full version [14].

### 3.2.3 Decomposition of Proximal Map

The computational cost of the proximal method depends strongly on the efficiency of solving the proximal step (8). A number of approaches have been developed for improving efficiency, including the minimum-norm-point approach [16] and the networkflow approach [17, 18]. Their computational efficiencies depend strongly on feature and partition size[2], however, which makes them inappropriate for our formulation because of potentially large feature and partition sizes.

Alternatively, this paper employs the decomposition of proximal maps [7]. The key idea here is to decompose the proximal step into a sequence of sub-problems that are easily solvable. We first introduce two easily-solvable proximal maps as follows:

$$M_1(B) = \underset{A}{\operatorname{argmin}} \frac{1}{2} \|A - B\|_F^2 + \eta^{(t)} \lambda_P \sum_{p=1}^{P} \|a_p\|_\infty, \quad (9)$$

$$M_2(B) = \underset{A}{\operatorname{argmin}} \frac{1}{2} \|A - B\|_F^2 + \eta^{(t)} \lambda_0 \sum_{p=0}^{P} \sum_{d=1}^{D} |a_{dp}|. \quad (10)$$

The theorem below guarantees that the decomposition of the proximal map (8) can be performed. The proof is provided in the full version.

**Theorem 1** *The original problem (8) can be decomposed into a sequence of two easily solvable proximal map problems as follows:*

$$A^{(t+1)} = M_0(A^{(t+\frac{1}{2})}) = M_2(M_1(A^{(t+\frac{1}{2})})). \quad (11)$$

The first proximal map (9) is the proximal operator with respect to the $L_{1,\infty}$-regularization. This problem can be decomposed into group-wise sub-problems. Each proximal operator with respect to each group can be computed through a projection on an $L_1$-norm ball (derived from the Moreau decomposition [16]), that is, $a_p = b_p - \operatorname*{argmin}_c \|c - b_p\|_2$ s.t. $\|c\|_1 \leq \eta^{(t)}\lambda$. This projection problem can be efficiently solved [19].

The second proximal map (10) is a well-known proximal operator with respect to $L_1$-regularization. This problem can be decomposed into element-wise ones and its solution is generated in a closed form through $a_{dp} = \operatorname{sgn}(b_{dp}) \max\left(0, |b_{dp} - \eta^{(t)}\lambda_0|\right)$. These two sub-problems can be easily solved, therefore, we can easily obtain the solution of the original proximal map (8).

$O(NP + \hat{P}D + PD \log D)$ is the computational complexity of partition-wise linear models where $\hat{P}$ is the number of active partitions. The procedure to derive the computational complexity, the implementation to speed up the optimization through warm start, and the summary of the iterative update procedure are written in the full version.

## 4 Generalization Bound Analysis

This section presents the derivation of a generalization error bound for partition-wise linear models and discusses how we can increase the number of partition candidates $P$ over the number of samples $N$. Our bound analysis is related to that of [20], which gives bounds for general overlapping group Lasso cases, while ours is specifically designed for partition-wise linear models.

Let us first derive an empirical Rademacher complexity [21] for a feasible weight space conditioned on (6). We can derive Rademacher complexity for our model using the Lemma below. Its proof is shown in the full version and this result is used to analyze the expected loss bound.

**Lemma 1** *If $\Omega(A) \leq 1$ is satisfied and if almost surely $\|x\|_\infty \leq 1$ with respect to $x \in \mathcal{X}$, the empirical Rademacher complexity for partition-wise linear models can be bounded as:*

$$\Re_A(X) = \frac{2^{3/2}}{\sqrt{N}}\left(2 + \sqrt{\ln(P + D(P+1))}\right) . \tag{12}$$

The next theorem shows the generalization bound of the global and local residual model. This bound is straightforwardly derived from Lemma 1 and the discussion of [21]. In [21], it has been shown that the uniform bound on the estimation error can be obtained through the upper bound of Rademacher complexity derived in Lemma 1. By using the uniform bound, the generalization bound of the global and local residual model defined in formula (4) can be derived.

**Theorem 2** *Let us define a set of weights that satisfies $\Omega_{group}(A) \leq 1$ as $\mathcal{A}$ where $\Omega_{group}(A)$ is as defined in Section 2.5 in [20]. Let a datum $(x_n, y_n)$ be i.i.d. sampled from a specific data distribution $\mathcal{D}$ and let us assume loss functions $\ell(\cdot, \cdot)$ to be L-Lipschitz functions with respect to a norm $\|\cdot\|$ and its range to be within $[0,1]$. Then, for any constant $\delta \in (0,1)$ and any $A \in \mathcal{A}$, the following inequality holds with probability at least $1 - \delta$.*

$$\mathbb{E}_{(x,y)\sim\mathcal{D}}\left[\ell(y, g(x))\right] \leq \frac{1}{N}\sum_{n=1}^{N}\ell(y_n, g(x_n)) + \Re_A(X) + \sqrt{\frac{\ln 1/\delta}{2N}} . \tag{13}$$

This theorem implies how we can increase the number of partition candidates. The third term of the right-hand side is obviously small if $N$ is large. The second term converges to zero with $N \to \infty$ if the value of $P$ is smaller than $o(e^N)$, which is sufficiently large in practice. In summary, we expect to handle a sufficient number of partition candidates for learning with little risk of over fitting.

## 5 Experiments

We conducted two types of experiments: 1) evaluation of how partition-wise linear models perform, on the basis of a simple synthetic dataset and 2) comparisons with state-of-the-art region-specific and locally linear models on the basis of standard classification and regression benchmark datasets.

### 5.1 Demonstration using Synthetic Dataset

We generated a synthetic binary classification dataset as follows. $x_n$s were uniformly sampled from a 20-dimensional input space in which each dimension had values between $[-1, 1]$. The target variables were determined using the XOR rule over the first and second features (the other 18 features

were added as noise for prediction purposes.), i.e., if the signs of first feature value and second feature value are the same, $y = 1$, otherwise $y = -1$. This is well known as a case in which linear models do not work. For example, $L_1$-regularized logistic regression produced nearly random outputs where the error rate was $0.421$.

We generated one partition for each feature except for the first feature. Each partition became active if the corresponding feature value was greater than $0.0$. Therefore, the number of candidate partitions was 19. We used the logistic regression function for loss functions. Hyper-parameters[3] were set as $\lambda_0 = 0.01$ and $\lambda_P = 0.001$. The algorithm was run in $1,000$ iterations.

Figure 2 illustrates results produced by the global and local residual model. The left-hand figure illustrates a learned effective partition (red line) to which the weight vector $a_1 = (10.96, 0.0, \cdots)$ was assigned. This weight $a_1$ was only applied to the region above the red line. By combining $a_1$ and the global weight $a_0$, we obtained the piece-wise linear representation shown in the right-hand figure. While it is yet difficult for existing piece-wise linear methods to capture global structures[4], our convex formulation makes it possible for the global and local residual model to easily capture the global XOR structures.

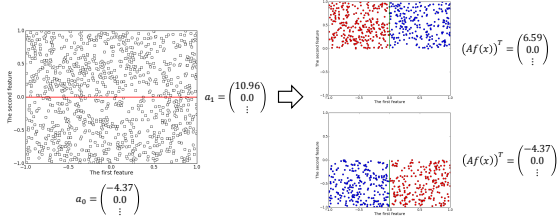

Figure 2: How the global and local residual model classifies XOR data. Red line indicates effective partition; green lines indicate local predictors; red circles indicate samples with $y = -1$; blue circles indicate samples with $y = 1$: This model classified XOR data precisely.

## 5.2 Comparisons using Benchmark Datasets

We next used benchmark datasets to compare our models with other state-of-the-art region-specific ones. In these experiments, we simply generated partition candidates (activeness functions) as follows. For continuous value features, we calculated all 5-quantiles for each feature and generated partitions at each quantile point. Partitions became active if a feature value was greater than the corresponding quantile value. For binary categorical features, we generated two partitions in which one became active when the feature was 1 (yes) and the other became active only when the feature value was 0 (no).

We utilized several standard benchmark datasets from UCI datasets (skin, winequality, census_income, twitter, internet_ad, energy_heat, energy_cool, communities), libsvm datasets (a1a, breast_cancer), and LIACC datasets (abalone, kinematics, puma8NH, bank8FM). Table 2 summarizes specifications for each dataset.

Table 2: Classification and regression datasets. $N$ is the size of data. $D$ is the number of dimensions. $P$ is the number of partitions. CL/RG denotes the type of dataset (CL: Classification/RG: Regression).

| | $N$ | $D$ | $P$ | CL/RG |
|---|---|---|---|---|
| skin | 245,057 | 3 | 12 | CL |
| winequality | 6,497 | 11 | 44 | CL |
| census_income | 45,222 | 105 | 99 | CL |
| twitter | 140,707 | 11 | 44 | CL |
| a1a | 1,605 | 113 | 452 | CL |
| breast-cancer | 683 | 10 | 40 | CL |
| internet_ad | 2,359 | 1,559 | 1,558 | CL |
| energy_heat | 768 | 8 | 32 | RG |
| energy_cool | 768 | 8 | 32 | RG |
| abalone | 4,177 | 10 | 40 | RG |
| kinematics | 8,192 | 8 | 32 | RG |
| puma8NH | 8,192 | 8 | 32 | RG |
| bank8FM | 8,192 | 8 | 32 | RG |
| communities | 1,994 | 101 | 404 | RG |

### 5.2.1 Classification

For classification, we compared the global and local residual model (Global/Local) with $L_1$ logistic regression (Linear), LSL-SP with linear discrimination analysis[5], LDKL supported by $L_2$-regularized hinge loss[6], FaLK-SVM with linear kernels[7], and C-SVM with RBF kernel[8]. Note that C-SVM is neither a region-specific nor a locally linear classification model; it is, rather, non-linear. We compared it with ours as a reference with respect to a common non-linear classification model.

Table 3: Classification results: error rate (standard deviation). The best performance figure in each dataset is denoted in **bold** typeface and the second best is denoted in ***bold italic***.

| | Linear | Global Local | LSL-SP | LDKL | FaLK-SVM | RBF-SVM |
|---|---|---|---|---|---|---|
| skin | 8.900 (0.174) | 0.249 (0.048) | 12.481 (8.729) | 1.858 (1.012) | **0.040** (0.016) | *0.229* (0.029) |
| winequality | 33.667 (1.988) | **23.713** (1.202) | 30.878 (1.783) | 36.795 (3.198) | 28.706 (1.298) | *23.898* (1.744) |
| census_income | 43.972 (0.404) | *35.697* (0.453) | **35.405** (1.179) | 47.229 (2.053) | – | 45.843 (0.772) |
| twitter | 6.964 (0.164) | *4.231* (0.090) | 8.370 (0.245) | 15.557 (11.393) | **4.135** (0.149) | 9.109 (0.160) |
| a1a | *16.563* (2.916) | **16.250** (2.219) | 20.438 (2.717) | 17.063 (1.855) | 18.125 (1.398) | 16.500 (1.346) |
| breast-cancer | 35.000 (4.402) | **3.529** (1.883) | *3.677* (2.110) | 35.000 (4.402) | – | 33.824 (4.313) |
| internet_ad | 7.319 (1.302) | **2.638** (1.003) | 6.383 (1.118) | 13.064 (3.601) | *3.362* (0.997) | 3.447 (0.772) |

Table 4: Regression results: root mean squared loss (standard deviation). The best performance figure in each dataset is denoted in **bold** typeface and the second best is denoted in ***bold italic***.

| | Linear | Global Local | RegTree | RBF-SVR |
|---|---|---|---|---|
| energy_heat | 0.480 (0.047) | *0.101* (0.014) | **0.050** (0.005) | 0.219 (0.017) |
| energy_cool | 0.501 (0.044) | **0.175** (0.018) | *0.200* (0.018) | 0.221 (0.026) |
| abalone | *0.687* (0.024) | **0.659** (0.023) | 0.727 (0.028) | 0.713 (0.025) |
| kinematics | 0.766 (0.019) | *0.634* (0.022) | 0.732 (0.031) | **0.347** (0.010) |
| puma8NH | 0.793 (0.023) | *0.601* (0.017) | 0.612 (0.024) | **0.571** (0.020) |
| bank8FM | 0.255 (0.012) | *0.218* (0.009) | 0.254 (0.008) | **0.202** (0.007) |
| communities | *0.586* (0.049) | **0.578** (0.040) | 0.653 (0.060) | 0.618 (0.053) |

For our models, we used logistic functions for loss functions. The max iteration number was set as 1000, and the algorithm stopped early when the gap in the empirical loss from the previous iteration became lower than $10^{-9}$ in 10 consecutive iterations. Hyperparameters[9] were optimized through 10-fold cross validation. We fixed the number of regions to 10 in LSL-SP, tree-depth to 3 in LDKL, and neighborhood size to 100 in FaLK-SVM.

Table 3 summarizes the classification errors. We observed 1) Global/Local consistently performed well and achieved the best error rates foir four datasets out of seven. 2) LSL-SP performed well for census_income and breast-cancer, but did significantly worse than Linear for skin, twitter, and a1a. Similarly, LDKL performed worse than Linear for census_income, twitter, a1a and internet_ad. This arose partly because of over fitting and partly because of bad local minima. Particularly noteworthy is that the standard deviations in LDKL were much larger than in the others, and the initialization issue would seem to become significant in practice. 3) FaLK-SVM performed well in most cases, but its computational cost was significantly higher than that of others, and it was unable to obtain results for census_income and internet_ad (we stopped the algorithm after 24 hours running).

### 5.2.2 Regression

For regression, we compared Global/Local with Linear, regression tree[10] by CART (RegTree) [1], and epsilon-SVR with RBF kernel[11]. Target variables were standardized so that their mean was 0 and their variance was 1. Performance was evaluated using the root mean squared loss in the test data. Tree-depth of RegTree and $\epsilon$ in RBF-SVR were determined by means of 10-fold cross validation. Other experimental settings were the same as those used in the classification tasks.

Table 4 summarizes RMSE values. In classification tasks, Global/Local consistently performed well. For the kinematics, RBF-SVR performed much better than Global/Local, but Global/Local was better than Linear and RegTree in many other datasets.

## 6 Conclusion

We have proposed here a novel convex formulation of region-specific linear models that we refer to as partition-wise linear models. Our approach simultaneously optimizes regions and predictors using sparsity-inducing structured penalties. For the purpose of efficiently solving the optimization problem, we have derived an efficient algorithm based on the decomposition of proximal maps. Thanks to its convexity, our method is free from initialization dependency, and a generalization error bound can be derived. Empirical results demonstrate the superiority of partition-wise linear models over other region-specific and locally linear models.

**Acknowledgments**

The majority of the work was done during the internship of the first author at the NEC central research laboratories.

## Footnotes

[2]For example, the fastest algorithm for the networkflow approach has $O(\mathcal{M}(B+1) \log(\mathcal{M}^2/(B+1)))$ time complexity, where $B$ is the number of breakpoints determined by the structure of the graph ($B \leq D(P+1) = O(DP)$) and $\mathcal{M}$ is the number of nodes, that is $P + D(P + 1) = O(DP)$ [17]. Therefore, the worst computational complexity is $O(D^2 P^2 \log DP)$.

[3]We conducted several experiments on other hyper-parameter settings and confirmed that variations in hyper-parameter settings did not significantly affect results.

[4]For example, a decision tree cannot be used to find a "true" XOR structure since marginal distributions on the first and second features cannot discriminate between positive and negative classes.

[5]The source code is provided by the author of [3].

[6]https://research.microsoft.com/en-us/um/people/manik/code/LDKL/download.html

[7]http://disi.unitn.it/~segata/FaLKM-lib/

[8]We used a libsvm package. http://www.csie.ntu.edu.tw/~cjlin/libsvm/

[9] $\lambda^1, \lambda_p^2$ in Global/Local, $\lambda^1$ in Linear, $\lambda_W, \lambda_\theta, \lambda_{\theta^{\cdot}}, \sigma$ in LDKL, $C$ in FaLK-SVM, and $C, \gamma$ in RBF-SVM.

[10] We used a scikit-learn package. `http://scikit-learn.org/`

[11] We used a libsvm package.

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
