[Reviews · NeurIPS 2014]

Submitted by Assigned_Reviewer_10

The paper introduces a novel convex region-specific linear models called partition-wise linear model. It assigns linear models to partitions of the input space and linear combination of these partition-specific models define the region-specific linear models. This allows them to construct convex objective functions. They optimize both the regions and predictors by using sparsity inducing structured penalties. As a specific example of the partition-wise linear model, they propose a global and local residual model which uses one global linear weight vector and a set of partition-wise linear weights. They provide generalization bounds from which they conclude that they can use a large number of partition candidates with little risk of over fitting. The algorithm for solving the optimization problem is based on the decomposition of proximal maps. They efficiently compare their method to others emphasizing that their method is the only one that achieves: joint optimization of regions and region specific predictors, initialization-independent optimizers, and meaningful generalization bounds. Experiments demonstrate the their models perform better or as well as region-specific or locally linear models.

Quality: Their claims are well supported both by theoretical analysis and experimental results. They carefully describe their new key ideas and derive generalization bounds. Thorough experiments show that their methods perform well in comparison with other methods.

Clarity: The paper is well-organized, but at times, it is not as clear as it could be.
This is primarily due to awkward sentence structure and missing articles.

Originality: The approach is a novel way to overcome the challenge of non-convex simultaneous optimization of region and region specific models. The related work is well referenced and effectively compared to their model.
Summary: Their new approach to addressing the challenge of non-convexity seems correctly derived and experimentally they show that their model performs better or as well as other models. The writing needs some improvement.

Submitted by Assigned_Reviewer_27

This paper address the problem of learning partition-wise linear models. Briefly, the data space is split into regions, and a linear model is defined in each region. By combining all those local linear models, a global nonlinear model is achieved. Previous learning algorithms typically suffer from local minimals due to non-convexity. This paper formulates the problem as "selecting" regions from a pre-defined region candidate list with sparsity-encouraging regularizers, resulting in a convex objective function. Efficient optimization procedures are derived based on decomposition of proximal maps, generalization bounds are given, and evaluation on benchmark datasets with comparison to related algorithms as well as standard nonlinear model baselines are used to show the effectiveness of the proposed algorithm.

This paper is well written and easy to follow. The proposed model is original and experiments shows that is is effective at solving relevant problems.
Summary: This paper solves a convex optimization problem to train partiton-wise linear models by selecting partitions with sparsity-inducing regularizers. Optimization procedures and generalization guarantee are given, and evaluation on standard benchmark datasets verifies the efficacy of the proposed model.

Submitted by Assigned_Reviewer_32

The paper proposes a new region-specific linear model, which is called partition-wise linear models. The idea of the proposed model which optimizes regions and predictors using sparsity-inducing structured penalties is interesting and the model works well compared to existing models in numerical experiments.

However, I have several concerns regarding the optimization techniques used in the paper.

First, the paper (line 200) mentions that (5) is an equivalent problem to (4). However, it is not true. Because (4) has a nonconvex constraint, \|a_p\|_0 \leq \mu_0, the resulting (5) has a duality gap even if lambda_0 is a Lagrange multiplier corresponding to the nonconvex constraint. Therefore, in general, the optimal solution of (5) does not coincide with that of (4).

Second, the convergence rates of the proximal gradient method and its accelerated method, discussed in Section 3.2.2, do not hold for the concerned problem (6).
To take advantage of these theoretical convergence rates, the loss function \ell should be differentiable. However, when the hinge loss is used for \ell (as written in line 147), the convergence speed becomes slower.
In addition, it is better to cite Nesterov (2007) instead of FISTA for the acceleration technique of the proximal gradient method.

Third, the paper mentions the "initialization independent" property of the proposed method as an advantage of the proposed model over existing models in Table 1.
But the initialization-independence is due to the convex relaxation (6) for the original problem (5). In my opinion, even though local optimization methods are initialization independent, the obtained local solution may be preferable to a relaxation solution unless the relaxation solution has a theoretical guarantee e.g., as the relaxation ratio.
Summary: The idea of the proposed model which optimizes regions and predictors using sparsity-inducing structured penalties is interesting. However, I have several concerns regarding the optimization techniques used in the paper (mainly, the equivalence of (4) and (5) and the convergence rates of the proposed method).
Author Feedback
Author rebuttal: We thank the reviewers for their comments and insightful reading of the paper. We will address all the reviewer's comments in the updated manuscript; we respond to the major concerns below.

***Assigned_Reviewer_32's comment***
First, the paper (line 200) mentions that (5) is an equivalent problem to (4). However, it is not true. Because (4) has a nonconvex constraint, \|a_p\|_0 \leq \mu_0, the resulting (5) has a duality gap even if lambda_0 is a Lagrange multiplier corresponding to the nonconvex constraint. Therefore, in general, the optimal solution of (5) does not coincide with that of (4).
---Answer---
Thank you for your helpful comment. We confirmed that was our careless misunderstanding and the equivalence between (4) and (5) is not satisfied.
Fortunately, this can be easily fixed with little change of the derivation process. First, we transform the constraint of the original problem (4) into the convex condition by deriving the tightest convex envelope of its constraint.
- min Σ l(y, g(x)) s.t. λ_P Σ_p | a_p|_∞ ≦ μ_P, λ_0 Σ_d | a_pd|_∞ ≦ μ_0 for all p
Then, we can transform this new problem to the equivalent problem (6), which can be proven to have the same optimal solution. We will reflect this modification in the updated manuscript.

***Assigned_Reviewer_32's comment***
Second, the convergence rates of the proximal gradient method and its accelerated method, discussed in Section 3.2.2, do not hold for the concerned problem (6).
To take advantage of these theoretical convergence rates, the loss function \ell should be differentiable. However, when the hinge loss is used for \ell (as written in line 147), the convergence speed becomes slower.
---Answer---
First, we have understood the point and therefore we denoted that "This achieves O(1/t) convergence [5] under Lipschitz-continuity of the loss gradient". We used the logistic loss as a differentiable loss for our model in the experimental section.
It becomes awkward to state hinge loss in line 147. We will state the squared hinge-loss instead of the hinge loss as a differential loss function in the updated manuscript.

***Assigned_Reviewer_32's comment***
In addition, it is better to cite Nesterov (2007) instead of FISTA for the acceleration technique of the proximal gradient method.
---Answer---
Thank you very much for your helpful comment.
We will cite Nesterov (2007) in the updated manuscript.

***Assigned_Reviewer_32's comment***
Third, the paper mentions the "initialization independent" property of the proposed method as an advantage of the proposed model over existing models in Table 1.
But the initialization-independence is due to the convex relaxation (6) for the original problem (5). In my opinion, even though local optimization methods are initialization independent, the obtained local solution may be preferable to a relaxation solution unless the relaxation solution has a theoretical guarantee e.g., as the relaxation ratio.
---Answer---
Thank you for the comment. First, although it is a good idea and interesting future direction to apply non-convex optimization like OMP rather than convex optimization, OMP-type optimization for overlapping group regularization problems has been yet unsolved as far as we know.
Second, it enables us to derive the theoretical guarantee of convergence speed and the upper bound of expected loss and we believe these are another good advantage to adopt convex formulation as well as the initialization-independence.